# Robot-Assisted Radical Prostatectomy for Locally Advanced Prostate Cancer: Oncological Potential and Limitations as the Primary Treatment

**DOI:** 10.3390/cancers17203286

**Published:** 2025-10-10

**Authors:** Noriyoshi Miura, Masaki Shimbo, Kensuke Shishido, Shota Nobumori, Naoya Sugihara, Takatora Sawada, Shunsuke Haga, Haruna Arai, Keigo Nishida, Osuke Arai, Tomoya Onishi, Ryuta Watanabe, Kenichi Nishimura, Tetsuya Fukumoto, Yuki Miyauchi, Tadahiko Kikugawa, Takato Nishino, Fumiyasu Endo, Kazunori Hattori, Takashi Saika

**Affiliations:** 1Department of Urology, Ehime University Graduate School of Medicine, Toon 791-0295, Japan; shishido.kensuke.ar@ehime-u.ac.jp (K.S.); nobumori.shota.ny@ehime-u.ac.jp (S.N.); sugihara.naoya.kp@ehime-u.ac.jp (N.S.); sawada.takatora.dx@ehime-u.ac.jp (T.S.); haga.shunsuke.xw@ehime-u.ac.jp (S.H.); arai.haruna.jl@ehime-u.ac.jp (H.A.); nishida.keigo.ir@ehime-u.ac.jp (K.N.); arai.osuke.gb@ehime-u.ac.jp (O.A.); onishi.tomoya.ey@ehime-u.ac.jp (T.O.); watanabe.ryuta.cu@ehime-u.ac.jp (R.W.); nishimura.kenichi.vx@ehime-u.ac.jp (K.N.); fukumoto.tetsuya.it@ehime-u.ac.jp (T.F.); miyauchi.yuki.mf@ehime-u.ac.jp (Y.M.); takikuga@m.ehime-u.ac.jp (T.K.); saika.takashi.ol@ehime-u.ac.jp (T.S.); 2Department of Urology, St. Luke’s International Hospital, Tokyo 104-8560, Japan; mashimbo@luke.ac.jp (M.S.); nisitaka@luke.ac.jp (T.N.); endofum@luke.ac.jp (F.E.); kazhat@luke.ac.jp (K.H.)

**Keywords:** prostate cancer, robot-assisted radical prostatectomy, biochemical recurrence-free survival, prostate-specific antigen, locally advanced

## Abstract

**Simple Summary:**

Prostate cancer is one of the most common cancers in men, and those with locally advanced disease are often treated with a combination of surgery, radiation, and systemic therapy. However, the long-term benefits of surgery alone are less clear. This study aimed to assess the safety and outcomes of robot-assisted radical prostatectomy (RARP) performed without additional systemic treatment. We analyzed 258 men with high-risk features—such as advanced clinical stage, high PSA levels, or aggressive biopsy results—who underwent RARP between 2012 and 2022. After a median follow-up of approximately 5 years, the 5-year survival rates were 36.6% for biochemical recurrence-free survival, 88.9% for metastasis-free survival, and 98.3% for cancer-specific survival. Factors linked to poorer outcomes included very high PSA, advanced stage, multiple high-grade biopsy cores, lymphovascular invasion, and lymph node involvement. These findings suggest that surgery alone may be effective for select patients, whereas multimodal therapy would be required in others.

**Abstract:**

*Background:* Locally advanced prostate cancer (PCa) is commonly treated with multimodal therapy; however, long-term outcomes of surgery alone are poorly defined. We investigated the potential and limitations of robot-assisted radical prostatectomy (RARP) as primary treatment without perioperative systemic therapy in patients with locally advanced PCa. *Methods:* We retrospectively analyzed 258 patients who underwent RARP with extended pelvic lymph node dissection between 2012 and 2022 with locally advanced PCa, defined as present if at least one of the following was met: clinical stage cT3b–T4; primary Gleason pattern 5; >4 biopsy cores with Grade Group 4 or 5; or more than one NCCN high-risk characteristic. Patients who received neoadjuvant or adjuvant therapy were excluded. Endpoints included biochemical recurrence-free survival, metastasis-free survival, cancer-specific survival, and predictors of persistent PSA. *Results:* Median follow-up was 60.6 months. Pathological stage ≥ pT3a occurred in 63.6% and nodal involvement (pN1) in 27.1%. Five-year BRFS, MFS, and CSS were 36.6%, 88.9%, and 98.3%, respectively. Persistent PSA occurred in 21.3%. Preoperative predictors included PSA > 40 ng/mL, clinical stage ≥ cT3a, and >4 biopsy cores with a Gleason score of 8–10; patients with ≥2 features had significantly poorer BRFS and MFS. Postoperative predictors of recurrence were pathological stage, lymphovascular invasion, and nodal involvement. *Conclusions:* RARP alone provided durable long-term cancer control in selected men with locally advanced PCa, whereas patients with multiple adverse features were unlikely to be cured with surgery alone. Careful risk stratification may identify candidates for surgical monotherapy and help avoid overtreatment, while others may benefit from multimodal therapy.

## 1. Introduction

Locally advanced prostate cancer (PCa) is a heterogeneous disease characterized by a high incidence of adverse pathological features, early biochemical recurrence (BCR), and an increased risk of cancer-specific mortality [1]. Current international guidelines generally recommend multimodal approaches, incorporating radical prostatectomy (RP), radiotherapy (RT), and/or systemic therapy, in order to optimize long-term oncological control [2,3,4].

Despite these recommendations, RP remains an important treatment option for select patients because it can provide durable local control, accurate pathological staging, and prognostic information that may guide subsequent therapy. With the widespread adoption of robot-assisted radical prostatectomy (RARP), surgical outcomes and perioperative safety have improved; however, most contemporary series assessing locally advanced or very high-risk (VHR) PCa include perioperative systemic or adjuvant therapies [5]. Consequently, the oncological efficacy of surgery alone in this population remains poorly understood.

Particularly, the long-term outcomes of RARP as the initial and sole treatment for patients with locally advanced PCa have not been sufficiently documented. Clarifying these outcomes is essential for improved understanding of the role of surgery in the modern multimodal era and for informed patient counseling and treatment selection.

This study aimed to evaluate the perioperative safety and long-term oncological outcomes of RARP without perioperative systemic therapy in patients with locally advanced prostate cancer and to investigate its potential and limitations as primary treatment. In addition, postoperative urinary incontinence was also evaluated. These factors are important indicators for measuring treatment feasibility and patient quality of life and are crucial when considering RARP as a standalone primary treatment option for locally advanced prostate cancer.

## 2. Materials and Methods

### 2.1. Patients

We retrospectively analyzed 1879 consecutive patients with prostate cancer who underwent RARP at our institution between July 2012 and November 2022. Locally advanced prostate cancer was defined, in according to the former NCCN very high-risk category, as present if at least one of the following was met: clinical stage cT3b–T4; primary Gleason pattern 5; >4 biopsy cores with Grade Group 4 or 5; or more than one NCCN high-risk characteristic. As older versions of the NCCN guidelines are not publicly available, we referred to the current NCCN Clinical Practice Guidelines in Oncology: Prostate Cancer (Version 2.2026), in which the modifications from the former definitions are described [2].

Among these patients, we excluded those who received perioperative systemic therapy (*n* = 24), did not undergo lymph node dissection (*n* = 15), and had incomplete clinical data (*n* = 8), leaving 258 patients for the final analysis. All included patients underwent RARP combined with extended pelvic lymph node dissection (ePLND) without neoadjuvant or adjuvant therapy. This study was approved by the Institutional Review Board of Ehime University Hospital (No. 1703014).

The preoperative and pathological data included age, PSA level at diagnosis, biopsy Gleason score (bGS), number of positive biopsy cores, clinical stage (cT), pathological stage (pT), pathological Gleason score (pGS), positive surgical margins (PSMs), lymphovascular invasion (LVI), number of nodes removed, and positive node status (pN1). Specimen confined disease was defined as cases with negative surgical margins and no lymph node metastasis (pN0). Perioperative outcomes included operative time, estimated blood loss, and 30-day postoperative complications classified according to the Clavien–Dindo classification.

### 2.2. Surgical Technique

All procedures were performed using the da Vinci Si or Xi system (Intuitive Surgical, Sunnyvale, CA, USA) through a six-port transperitoneal approach. Expanded RARP was performed using an extrafascial approach, extending the dissection to the anterior rectal wall and elevating the perirectal fat and Denonvilliers’ fascia with the specimen [6,7]. This method enables the complete removal of perirectal fat along with the prostate. ePLND included lymph nodes above the external iliac axis, within the obturator fossa and around the internal iliac artery [8].

### 2.3. Follow-Up

None of the patients received neoadjuvant or adjuvant therapy. PSA levels were monitored every 3 months during the first year and semi-annually thereafter. Biochemical recurrence (BCR) was defined as two consecutive PSA values ≥ 0.2 ng/mL after postoperative PSA had decreased below 0.1 ng/mL consistent with contemporary guideline definitions [9]. Persistent PSA was defined as a postoperative PSA that did not decline to <0.1 ng/mL after radical prostatectomy; for time-to-event analyses, patients with Persistent PSA were counted as events at time zero for BRFS, with the date of surgery assigned as the BCR date [10]. Salvage therapy (sRT and/or ADT) was considered only after biochemical recurrence (BCR), consistent with our definitions. The choice of sRT, ADT, or combined ADT + RT was individualized based on PSA kinetics, imaging findings, pathological risk factors, and patient preference. Clinical recurrence was defined as positive imaging results after BCR. Urinary continence (UC) recovery was defined as the use of zero or one safety pad at the last follow-up.

### 2.4. Statistical Analysis

The study endpoints were biochemical recurrence-free survival (BRFS), metastasis-free survival (MFS), and cancer-specific survival (CSS), measured from the date of surgery to the event or last follow-up. Logistic regression was used to identify the preoperative risk factors for persistent PSA. Multivariate Cox regression analyses were performed to evaluate the predictors of BCR and MFS. The covariates included PSA level, pathological T stage, a pathological Gleason score ≥8, primary Gleason grade 5, LVI, lymph node invasion, and PSMs.

Baseline characteristics were compared using the χ^2^ test for categorical variables and the Kruskal–Wallis test for continuous variables. Survival curves were estimated using the Kaplan–Meier method, and differences were assessed using the log-rank test. Statistical analyses were conducted using EZR [11], and statistical significance was set at *p* < 0.05.

## 3. Results

### 3.1. Patient Characteristics

The median follow-up was 60.6 months (IQR, 38.8–87.3). More than 90% of patients had a Gleason Grade Group ≥ 4, and primary Gleason pattern 5 was observed in 31.4%. Clinical stage ≥ cT3a was present in 57.9% of patients (Table 1).

### 3.2. Perioperative Outcomes

Perioperative complications occurred in 91 of 259 patients (35.1%), most of which were Grade ≤2. Grade 3a–b complications were observed in 27 patients (10.4%). The most frequent complications were inguinal hernias (*n* = 13) and wound hernias (*n* = 4). One case of vesicorectal fistula occurred after an intraoperative rectal injury (Table 2).

### 3.3. Pathological Outcomes

The median number of removed lymph nodes was 19 (IQR, 13–26). Pathological stage pT3a was observed in 63.6% of patients. Positive surgical margins were present in 37.6%, and lymph node metastases (pN1) in 27.1%. Specimen-confined disease was achieved in 50.0% of cases (Table 3).

### 3.4. Oncological Outcomes

The median biochemical recurrence (BCR) survival was 33.5 months. The 5-year and 8-year biochemical recurrence-free survival (BRFS) rates were 36.6% and 33.9%, respectively. Persistent PSA occurred in 55 (21.3%) patients. The 5-year metastasis-free and cancer-specific survival (MFS) rates were 88.9%, and the 5-year cancer-specific survival (CSS) rate was 98.3%. Among patients who developed BCR, subsequent treatments were initiated according to disease status. Salvage radiotherapy (sRT) was performed in 23 patients, androgen deprivation therapy (ADT) in 75 patients, and a combination of ADT and RT in 40 patients. Additionally, 16 patients with biochemical recurrence but persistently low PSA levels were managed with active surveillance without immediate treatment. Patients who experienced neither Persistent PSA nor BCR continued scheduled PSA surveillance (Figure 1).

### 3.5. Prognostic Factors

The results of multivariable analysis showed that preoperative PSA, clinical stage cT3a/b, a Gleason Grade Group ≥ 4, and >4 biopsy cores with a Gleason score of 8–10 were significant predictors of PSA (Table 4). Kaplan–Meier analysis showed that patients with ≥2 of these risk factors had significantly worse BRFS (*p* < 0.001) and MFS (*p* = 0.008), compared with those having ≤1 factor (Figure 2).

Postoperatively, pathological T stage, lymphovascular invasion (LVI), and pN1 status were independent predictors of BCR (Table 5). Additionally, LVI positivity and persistent PSA were significant predictors of MFS (Table 6).

### 3.6. Functional Outcomes

Among 187 patients with available postoperative UC data, 173 (92.5%) recovered from urinary continence within 12 months (Figure 3).

## 4. Discussion

This study evaluated the perioperative, pathological, and oncological outcomes of RARP without perioperative systemic therapy in patients with locally advanced prostate cancer (PCa). More than 90% of patients had a Gleason Grade Group ≥ 4, and nearly one-third harbored primary Gleason pattern 5, underscoring the aggressive nature of this cohort. Despite these adverse features, RARP with extended lymph node dissection (ePLND) has achieved excellent long-term cancer control in a substantial proportion of men, with a 5-year cancer-specific survival (CSS) rate of 98.3% and a metastasis-free survival (MFS) rate of 88.9%.

Perioperative morbidity was acceptable, with most complications being minor (Clavien–Dindo grade ≤ 2) and severe complications in only 10.4% of patients. These findings support the feasibility of surgical monotherapy in carefully selected cases, when performed in high-volume, experienced centers.

From a pathological perspective, more than 60% of patients were upstaged to ≥pT3a disease and 27% had nodal involvement (pN1), highlighting the aggressive biology of this population. Nevertheless, approximately half of the patients achieved specimen-confined disease, which was strongly associated with favorable outcomes, indicating that cure with surgery alone is possible in a subset of locally advanced cases.

Although biochemical recurrence-free survival (BRFS) was limited (5-year BRFS, 36.6%), long-term metastasis control and cancer-specific survival (CSS) rates remained high. Importantly, we identified the prognostic factors that may help distinguish patients who are unlikely to benefit from surgery alone. Preoperatively, PSA > 40 ng/mL, clinical stage ≥cT3a, and >4 biopsy cores with a Gleason score of 8–10 independently predicted persistent PSA. Patients with ≥2 of these features had significantly worse BRFS and MFS, suggesting that they should be counseled toward multimodal treatment. Conversely, patients without persistent PSA demonstrated a plateau in BRFS and an excellent MFS/CSS, supporting the role of surgery as an adequate monotherapy in selected cases.

Postoperatively, the pathological stage, lymphovascular invasion (LVI), and nodal involvement (pN1) were independent predictors of BCR. Our findings corroborate those of Ozawa et al. [12], who demonstrated that LVI independently predicts BCR and is associated with nodal positivity in a large RARP cohort. Furthermore, LVI positivity and persistent PSA were significant predictors of MFS, underscoring their importance in risk stratification and postoperative surveillance.

In our study, patients who had received systemic therapy or radiotherapy as neoadjuvant or adjuvant treatment were excluded from the analysis. Therefore, all included patients underwent RARP with extended pelvic lymph node dissection alone. Consequently, all patients who received radiotherapy in this cohort underwent it as salvage radiotherapy following biochemical recurrence. The ARTISTIC meta-analysis, which prospectively integrated data from three randomized trials—RADICALS-RT, GETUG-AFU 17, and RAVES—demonstrated no significant difference in event-free survival between adjuvant radiotherapy and early sRT (5-year event-free survival: 89% vs. 88%; HR 0.95, 95% CI 0.75–1.21, *p* = 0.70) [13]. These findings indicate that immediate adjuvant RT does not necessarily provide additional benefit and support the validity of a management policy where adjuvant RT is deferred until recurrence. In our cohort, most patients who experienced recurrence received appropriate salvage therapies, while some with very low PSA levels were safely monitored without immediate intervention. Taken together, these results suggest that the omission of adjuvant RT in our study should not be regarded as undertreatment, but rather reflects a treatment approach consistent with current high-level evidence.

RT combined with androgen deprivation therapy (ADT) remains the guideline-recommended standard for locally advanced PCa [2,3,4]. Nevertheless, RP continues to offer unique advantages such as accurate pathological staging, potential deferral of systemic therapy, and the possibility of durable disease control in selected patients. Several comparative studies have reported similar CSS and OS rates between RP and RT, after adjusting for patient selection [14,15]. However, the true comparative benefit of RP as part of multimodal therapy versus primary EBRT plus ADT remains unclear. The ongoing SPCG-15 randomized trial is expected to clarify this issue by directly comparing RP (with or without adjuvant or salvage EBRT) with EBRT plus ADT in patients with locally advanced disease [16]. In the absence of randomized data, the current evidence relies mainly on retrospective series and meta-analyses, which are inherently limited by selection bias. While some studies suggested a survival advantage for RP, others reported equivalence when modern multimodal RT strategies, including EBRT plus brachytherapy and ADT, were applied [17].

A concern of pelvic RT is the risk of second primary malignancies. A systematic review and meta-analysis published in the BMJ demonstrated that RT for prostate cancer was associated with significantly increased risks of bladder, colorectal, and rectal cancers, compared with patients treated surgically or without RT, although absolute risks remained low [18]. Consistent with this, a large contemporary veterans’ affairs cohort study also showed that men who received primary RT had a significantly higher incidence of secondary cancers than those treated surgically, with the relative risk increasing steadily beyond 10–15 years of follow-up [19]. Taken together, these findings suggest that RT carries a higher long-term risk of secondary malignancies than surgery. Therefore, this potential late adverse effect should be carefully considered during shared decision making, particularly in younger patients with longer life expectancies. A distinct late concern is second primary malignancy after pelvic RT.

The unique contribution of this study lies in its exclusive focus on surgical monotherapy. Unlike most contemporary studies that combine surgery with planned adjuvant or early salvage therapy, our cohort received salvage treatment only upon BCR. This “surgery-first” strategy provides a benchmark of intrinsic surgical efficacy and helps disentangle the effect of RP itself from that of subsequent multimodal therapy.

These findings have important clinical implications. Patients with multiple adverse preoperative features (≥cT3a, PSA > 40 ng/mL, and multiple GS 8–10 biopsy cores) are unlikely to be controlled with surgery alone and should be counseled toward multimodal approaches. In contrast, patients without persistent PSA achieved durable long-term control, reinforcing the value of RARP monotherapy in select men and highlighting the need to avoid overtreatment.

The limitations of this study included its retrospective design and potential selection bias inherent to the surgical series. Furthermore, although the median follow-up period exceeded five years, long-term data are necessary to fully assess cancer-specific mortality. Future prospective trials should validate risk-adapted multimodal strategies and incorporate molecular and genomic classifiers to refine patient selection, particularly for those with multiple high-risk features.

## 5. Conclusions

In 258 patients with locally advanced prostate cancer treated with RARP alone, perioperative morbidity was acceptable, and long-term cancer control was favorable, with a 5-year MFS of 88.9% and CSS of 98.3%. Both preoperative factors (≥cT3a, PSA > 40 ng/mL, multiple Gleason Grade Group 4–5 cores) and postoperative features (pathological stage, LVI, pN1) independently predicted recurrence.

These findings indicate that although RARP monotherapy can achieve durable long-term outcomes in selected patients, those harboring multiple adverse features are unlikely to be cured with surgery alone and should be considered for multimodal therapy. Careful risk stratification is therefore essential to identify candidates for surgical monotherapy and guide the integration of additional systemic therapy or RT when appropriate.

## Figures and Tables

**Figure 1 cancers-17-03286-f001:**
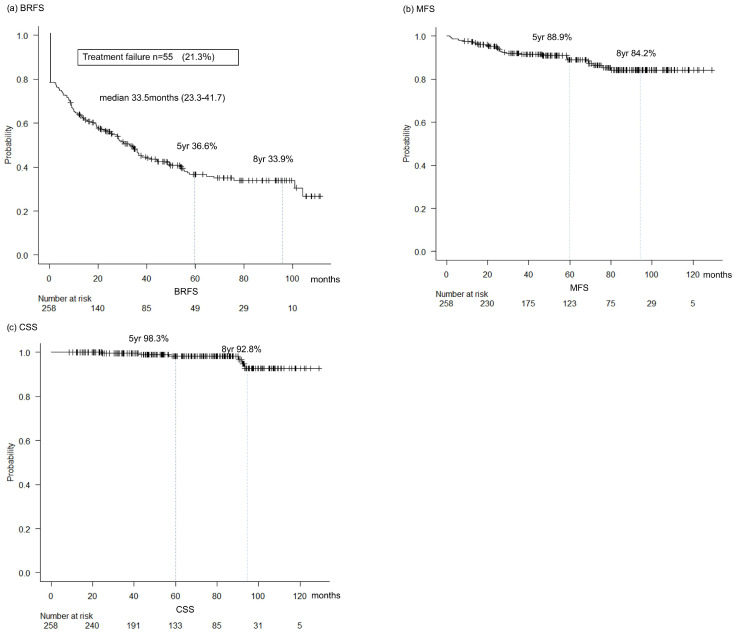
Kaplan–Meier curves for (**a**) biochemical recurrence-free survival (BRFS), (**b**) metastasis-free survival (MFS) and (**c**) cancer-specific survival.

**Figure 2 cancers-17-03286-f002:**
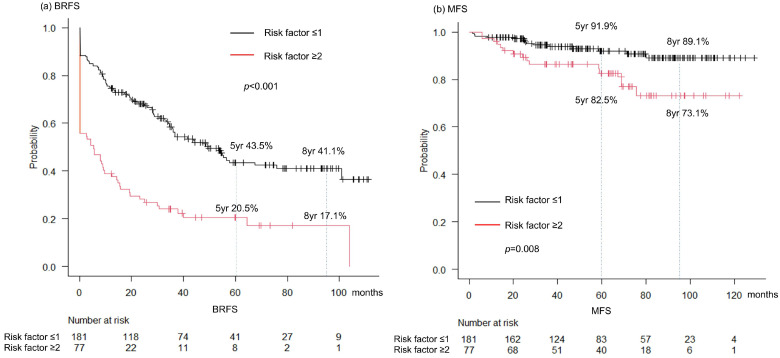
Kaplan–Meier curves for (**a**) biochemical recurrence-free survival (BRFS) and (**b**) metastasis-free survival (MFS) stratified by number of adverse preoperative risk factors (≥T3a, PSA > 40, biopsy cores with GS 8–10).

**Figure 3 cancers-17-03286-f003:**
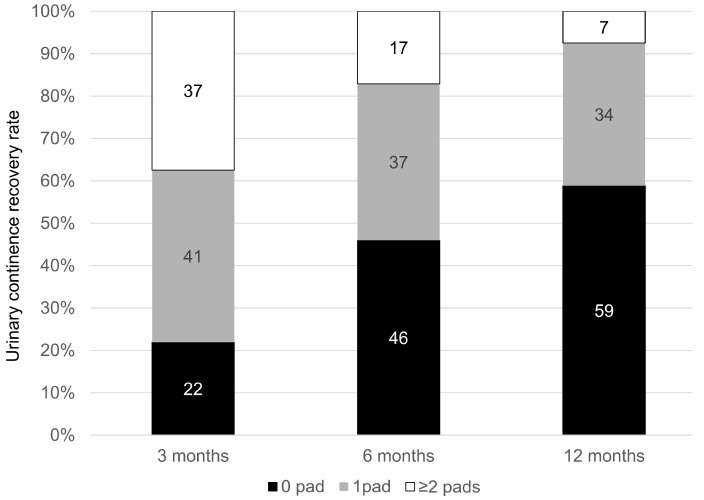
Subgroup analysis of urinary continence recovery rate in 187 patients (72.6%) who had complete postoperative urinary continence data. One-hundred seventy-three patients (92.5%) recovered their urinary continence within 12 months. Definition of urinary continence: 0–1 pad/day.

**Table 1 cancers-17-03286-t001:** Patients’ characteristics. IQR: interquartile range.

Patients Characteristics	*n* = 258
Age (yr)/median (IQR)	69 (65–73)
PSA (ng/mL) median (IQR)	14.6 (7.3–27.4)
Biopsy ISUP Gleason Grading (%)	
1	2 (0.8)
2	9 (3.5)
3	7 (2.7)
4	77 (29.8)
5	163 (63.2)
Primary Gleason grade 5 (%)	81 (31.4)
>4 biopsy cores with GS 8–10 (%)	113 (46.1)
Clinical stage (%)	
≤cT2	109 (42.1)
cT3a	113 (43.6)
cT3b	37 (14.3)
cT4	0 (0)
Follow up period (months)/median (IQR)	60.6 (38.8–87.3)

**Table 2 cancers-17-03286-t002:** Perioperative outcomes of prostate cancer patients with locally advanced prostate cancer treated with extended robot-assisted laparoscopic prostatectomy.

Perioperative Outcomes	*n* = 258
Operative time (min) median (IQR)	288 (239–338)
Console time (min) median (IQR)	237 (189–280)
Blood loss (mL) median (IQR)	100 (10–200)
Postoperative complications (%)	91 (35.1%)
Clavien-Dindo classification (%)	
I	25 (9.7%)
II	39 (15.1%)
IIIa	9 (3.5%)
IIIb	18 (6.9%)
IV	0 (0%)
V	0 (0%)
Postoperative complications	
IIIa Urethral stenosis	1
Ileus	1
Wound hernia	4
Infection	2
Vesicorectal fistula	1
IIIb Inguinal hernia	13
Urethral stenosis	2
Infection of lymphocele	2
Wound hernia	1

**Table 3 cancers-17-03286-t003:** Pathological outcomes.

Pathological Outcomes	*n* = 258
Pathological stage (%)	
≤pT2	94 (36.4)
pT3a	87 (33.7)
pT3b	76 (29.5)
pT4	1 (0.4)
Pathological Gleason score (%)	
1	0 (0)
2	17 (6.6)
3	29 (11.2)
4	58 (22.5)
5	154 (59.7)
Lymphovascular invasion (%)	154 (65.5)
Positive surgical margins (%)	97 (37.6)
pN1	70 (27.1)
Number of nodes removed /median (IQR)	19 (13–26)
Number of positive nodes /median (IQR)	1 (1–3)
Specimen confined disease (%)	129 (50.0)

**Table 4 cancers-17-03286-t004:** Univariate and multivariable logistic regression analyses assessing postoperative predictors of persistent PSA in prostate cancer patients with locally advanced prostate cancer treated with robot-assisted radical prostatectomy.

Baseline Characteristics	Univariable Analysis			Multivariable Analysis		
	OR	95%CI	*p*-Value	HR	95%CI	*p*-Value
PSA at diagnosis						
<10	1.00			1.00		
10–20	1.46	0.62–3.42	0.384	1.50	0.57–3.96	0.413
20–40	1.53	0.69–3.42	0.300	1.88	0.74–4.74	0.181
40<	4.00	1.65–9.72	0.002	4.91	1.69–14.2	0.0034
cT stage ≤ T2a	1.00			1.00		
T2b-c	0.96	0.28–3.37	0.953	0.77	0.21–2.82	0.690
T3a	2.85	1.11–7.36	0.030	2.80	1.00–7.81	0.049
T3b	6.60	2.27–19.2	<0.001	8.61	2.61–28.4	<0.001
GG ≥ 4	4.94	0.64–37.9	0.125	9.91	1.12–87.6	0.039
>4 cores of GG 4	2.91	1.52–5.57	<0.001	2.88	1.38–6.04	0.005
Primary Gleason grade 5	0.69	0.35–1.36	0.29	0.99	0.45–2.21	0.987

**Table 5 cancers-17-03286-t005:** Univariate and multivariable Cox regression analyses assessing postoperative predictors of biochemical recurrence in prostate cancer patients with locally advanced prostate cancer treated with robot-assisted radical prostatectomy.

Baseline Characteristics	Univariable Analysis			Multivariable Analysis		
	HR	95%CI	*p*-Value	HR	95%CI	*p*-Value
PSA at diagnosis						
<10	1.00			1.00		
10–20	1.36	0.90–2.06	0.15	1.03	0.66–1.62	0.889
20–40	0.96	0.63–1.45	0.84	0.85	0.55–1.32	0.468
40<	1.66	1.02–2.72	0.043	1.19	0.69–2.05	0.530
pT stage T2	1.00			1.00		
T3a	2.03	1.34–3.08	<0.001	1.64	1.04–2.61	0.035
T3b	3.29	2.18–4.99	<0.001	2.12	1.29–3.46	0.003
T4	9.65	1.30–71.4	0.026	4.94	0.63–39.0	0.130
Prostatectomy GG ≥ 4	1.50	0.95–2.38	0.085	1.05	0.64–1.70	0.856
LVI positive	2.68	1.80–3.98	<0.001	1.72	1.10–2.69	0.017
Resection margin positive	1.41	1.02–1.95	0.038	1.02	0.71–1.48	0.900
pN1 (vs. pN0)	2.92	2.09–4.08	<0.001	1.79	1.20–2.67	0.005

**Table 6 cancers-17-03286-t006:** Univariate and multivariable Cox regression analysis assessing postoperative predictors of metastasis-free survival in prostate cancer patients with locally advanced prostate cancer treated with robot-assisted radical prostatectomy.

Baseline Characteristics	Univariable Analysis			Multivariable Analysis		
	HR	95%CI	*p*-Value	HR	95%CI	*p*-Value
PSA at diagnosis						
<10	1.00			1.00		
10–20	0.80	0.30–2.14	0.657	0.46	0.15–1.38	0.165
20–40	0.53	0.19–1.51	0.233	0.34	0.10–1.13	0.078
40<	1.40	0.52–3.73	0.503	0.47	0.15–1.48	0.197
pT stage T2	1.00			1.00		
T3a	3.13	1.00–9.83	0.051	2.35	0.61–8.97	0.212
T3b	4.63	1.52–14.1	0.007	2.83	0.73–11.0	0.133
T4	NA	0-NA	0.998	NA	0.00-NA	0.998
Prostatectomy GG ≥ 4	5.33	0.73–39.2	0.100	2.37	0.31–17.9	0.403
LVI positive	7.79	1.83–33.2	0.006	5.00	1.12–22.3	0.035
Resection margin positive	1.43	0.68–2.97	0.343	0.84	0.34–2.06	0.697
pN1 (vs pN0)	1.43	0.66–3.08	0.361	0.50	0.20–1.23	0.130
Persistent PSA	5.97	2.85–12.5	<0.001	5.66	2.29–14.0	<0.001

## Data Availability

The raw data supporting the conclusions of this study will be made available upon request.

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
