# Peer review of "Robot-Assisted Radical Prostatectomy for Locally Advanced Prostate Cancer: Oncological Potential and Limitations as the Primary Treatment"

_cancers, 2025, doi:10.3390/cancers17203286_

Round 1
Reviewer 1 Report
Comments and Suggestions for Authors
Comments to the authors
General comments
This article examined the treatment outcomes and perioperative safety of RARP as monotherapy for locally advanced prostate cancer without preoperative or adjuvant therapy. The high-risk case group studied in this article is intriguing, but the paper's content—including introduction, methods, and results—revealed numerous points that are inadequately documented. Based on the content described, it must be concluded that the presentation of this research article is inadequate.
Specific comments
Major
- This article evaluated MFS and CSS in addition to BCR, but lacks information regarding salvage therapy. How many SRT and SHT procedures were performed? Or were none performed at all? If SRT or SHT was performed, documentation regarding this must be included in the Material and Methods and Results sections.
- In the Introduction, the purpose of this article is stated as perioperative safety. However, there is no mention of why perioperative complications and urinary status were evaluated. Without demonstrating some basis for the deterioration of perioperative complications or postoperative urinary status in high-risk prostate cancer treatment, the objective of this article remains unsubstantiated.
- The section on Materials and Methods included descriptions based on the former NCCN guidelines; however, references are required for this information.
4) Although it is described as “treatment failure,” it is generally referred to as “persistent PSA.”
5) The statement “Gleason score > 8” in the “Statistical analysis” section of the “Materials and Methods” may be corrected to “Gleason score ≥ 8.”
Minor
- What is the definition of Specimen confined disease?
- Table4, Figure2, Table5, Table6. The notation for ≥ is incorrect. (≧is not correct for English literature.)
Author Response
I would like to express my sincere gratitude to the reviewers for their thoughtful comments and for providing me with the opportunity to clarify several important points regarding this research.
Major
Comment 1:
This article evaluated MFS and CSS in addition to BCR, but lacks information regarding salvage therapy. How many SRT and SHT procedures were performed? Or were none performed at all? If SRT or SHT was performed, documentation regarding this must be included in the Material and Methods and Results sections.
Response
We thank the reviewer for this important comment. We did not administer any adjuvant therapy; however, salvage therapy was performed. Specifically, salvage radiotherapy (sRT) was delivered in 23 patients, androgen deprivation therapy (ADT) in 75 patients, and combined ADT+RT in 40 patients. In addition, 16 patients who developed biochemical recurrence (BCR) but had persistently low PSA levels were managed with active surveillance without immediate treatment.
We have now added this information to Materials and Methods section and Results section for clarity as below.
Materials and Methods section (insert)
“Salvage therapy (sRT and/or ADT) was considered only after biochemical recurrence (BCR), consistent with our definitions. The choice of sRT, ADT, or combined ADT+RT was individualized based on PSA kinetics, imaging findings, pathological risk factors, and patient preference.” (Page 3, Line 114-117)
Results section (insert)
“Among patients who developed BCR, subsequent treatments were initiated according to disease status. Salvage radiotherapy (sRT) was performed in 23 patients, androgen deprivation therapy (ADT) in 75 patients, and a combination of ADT and RT in 40 patients. Additionally, 16 patients with biochemical recurrence but persistently low PSA levels were managed with active surveillance without immediate treatment. Patients who experienced neither Persistent PSA nor BCR continued scheduled PSA surveillance.” (Page 5, Line 158- Page 6 Line 164)
Comment 2:
In the Introduction, the purpose of this article is stated as perioperative safety. However, there is no mention of why perioperative complications and urinary status were evaluated. Without demonstrating some basis for the deterioration of perioperative complications or postoperative urinary status in high-risk prostate cancer treatment, the objective of this article remains unsubstantiated.
Response:
Thank you for this valuable comment. We agree that the rationale was not sufficiently explicit. We have revised the Introduction to clarify why perioperative complications and postoperative urinary status were evaluated:
“This study aimed to evaluate the perioperative safety and long-term oncological outcomes of RARP without perioperative systemic therapy in patients with locally advanced prostate cancer, and to investigate its potential and limitations as primary treatment. In addition, postoperative urinary incontinence was also evaluated. These factors are important indicators for measuring treatment feasibility and patient quality of life and are crucial when considering RARP as a standalone primary treatment option for locally advanced prostate cancer.” (Page 2, Line 68-72)
Comment 3:
The section on Materials and Methods included descriptions based on the former NCCN guidelines; however, references are required for this information.
Response:
Thank you very much for your valuable comment. We agree that the description of the criteria for defining locally advanced prostate cancer should be supported by a reference. Since the older versions of the NCCN guidelines are not publicly available, we referred to the current NCCN Clinical Practice Guidelines in Oncology: Prostate Cancer (Version 2.2026), which explicitly describes the modifications from the former definitions. We have therefore added this reference in the Materials and Methods section to clarify the source.
Revised text (Materials and Methods):
“Locally advanced prostate cancer was defined, according to the former NCCN guidelines for very high-risk disease, as meeting at least two of the following criteria: clinical stage ≥cT3, prostate-specific antigen (PSA) ≥40 ng/mL, and Gleason Grade Group ≥4. As older versions of the NCCN guidelines are not publicly available, we referred to the current NCCN Clinical Practice Guidelines in Oncology: Prostate Cancer (Version 2.2026), in which the modifications from the former definitions are described 【2】.” (Page 2, Line79-82)
Comment 4:
Although it is described as “treatment failure,” it is generally referred to as “persistent PSA.”
Response:
Thank you for catching this. We agree that “persistent PSA” is the more precise and conventionally used term in prostate cancer postoperative monitoring. Accordingly, we have revised the manuscript and replaced every instance of “treatment failure” with “persistent PSA,” including in the definition, outcome description, statistical analyses, and results.
Revised text:
Abstract
Endpoints included biochemical recurrence–free survival (BRFS), metastasis-free survival (MFS), cancer-specific survival (CSS), and predictors of treatment failure.
→Endpoints included biochemical recurrence–free survival (BRFS), metastasis-free survival (MFS), cancer-specific survival (CSS), and predictors of persistent PSA. (Page1 Line 34)
Treatment failure occurred in 21.3%. Preoperative predictors included PSA >40 ng/mL, clinical stage ≥cT3a, and >4 biopsy cores with Gleason score 8–10; patients with ≥2 features had signifi-cantly poorer BRFS and MFS.
→Persistent PSA occurred in 21.3%. Preoperative predictors included PSA >40 ng/mL, clinical stage ≥cT3a, and >4 biopsy cores with Gleason score 8–10; patients with ≥2 features had significantly poorer BRFS and MFS. (Page 1, Line 36)
Material and methods
“Treatment failure was defined as a PSA level not decreasing to <0.1 ng/mL after surgery, with the date of surgery assigned as the BCR date.”
→“Persistent PSA was defined as a PSA level not decreasing to <0.1 ng/mL after surgery, with the date of surgery assigned as the BCR date.” (Page3 Line 111)
Logistic regression was used to identify the preoperative risk factors for treatment failure.
→Logistic regression was used to identify the preoperative risk factors for persistent PSA. (Page3 Line 124)
Treatment failure occurred in 55 (21.3%) patients.
→Persistent PSA occurred in 55 (21.3%) patients. (Page5 Line156)
The results of multivariable analysis showed that preoperative PSA, clinical stage cT3a/b, Gleason Grade Group ≥4, and >4 biopsy cores with Gleason score 8–10 were significant predictors of treatment failure (Table 4).
→The results of multivariable analysis showed that preoperative PSA, clinical stage cT3a/b, Gleason Grade Group ≥4, and >4 biopsy cores with Gleason score 8–10 were significant predictors of persistent PSA (Table 4). (Page 6 Line185)
Table 4. Univariate and multivariable logistic regression analyses assessing postoperative predictors of treatment failure in prostate cancer patients with locally advanced prostate cancer treated with robot-assisted radical prostatectomy.
→Table 4. Univariate and multivariable logistic regression analyses assessing postoperative predictors of persistent PSA in prostate cancer patients with locally advanced prostate cancer treated with robot-assisted radical prostatectomy. (Page 6 Line189)
Additionally, LVI positivity and treatment failure were significant predictors of MFS (Table 6).
→Additionally, LVI positivity and persistent PSA were significant predictors of MFS (Table 6).(Page7 Line198)
Table 6 Treatment failure → Persistent PSA (Page 8 Table 6)
Discussion
Preoperatively, PSA >40 ng/mL, clinical stage ≥cT3a, and >4 biopsy cores with Gleason score 8–10 independently predicted treatment failure persistent PSA. Patients with ≥2 of these features had significantly worse BRFS and MFS, suggesting that they should be counseled toward multimodal treatment. Conversely, patients without treatment failure persistent PSA demonstrated a plateau in BRFS and an excellent MFS/CSS, supporting the role of surgery as an adequate monotherapy in selected cases
→Preoperatively, PSA >40 ng/mL, clinical stage ≥cT3a, and >4 biopsy cores with Gleason score 8–10 independently predicted persistent PSA. Patients with ≥2 of these features had significantly worse BRFS and MFS, suggesting that they should be counseled toward multimodal treatment. Conversely, patients without persistent PSA demonstrated a plateau in BRFS and an excellent MFS/CSS, supporting the role of surgery as an adequate monotherapy in selected cases. (Page9, Line234-240)
Furthermore, LVI positivity and treatment failure were significant pre-dictors of MFS, underscoring their importance in risk stratification and postoperative surveillance.
→Furthermore, LVI positivity and persistent PSA were significant pre-dictors of MFS, underscoring their importance in risk stratification and postoperative surveillance.(Page10 Line245)
In contrast, patients without treatment failure achieved durable long-term control, reinforcing the value of RARP monotherapy in select men and highlighting the need to avoid overtreatment.
→In contrast, patients without persistent PSA achieved durable long-term control, reinforcing the value of RARP monotherapy in select men and highlighting the need to avoid overtreatment.(Page11 Line297)
Comment 5:
The statement “Gleason score > 8” in the “Statistical analysis” section of the “Materials and Methods” may be corrected to “Gleason score ≥ 8.”
Response:
We appreciate the reviewer’s careful observation. We agree with this correction and have revised the text accordingly in the Materials and Methods section.
Multivariate Cox regression analyses were performed to evaluate the predictors of BCR and MFS. The covariates included PSA level, pathological T stage, pathological Gleason score >8, primary Gleason grade 5, LVI, lymph node invasion, and PSMs.
→Multivariate Cox regression analyses were performed to evaluate the predictors of BCR and MFS. The covariates included PSA level, pathological T stage, pathological Gleason score ≥8, primary Gleason grade 5, LVI, lymph node invasion, and PSMs. (Page 3, Line126)
Minor
Comment 1:
What is the definition of “specimen confined disease”?
Response:
Thank you for your valuable comment. We have clarified the definition in the Materials and Methods section. In our study, specimen confined disease was defined as cases with negative surgical margins and no lymph node metastasis (pN0). We have revised the Materials and Methods section as follows (new text in bold).
The preoperative and pathological data included age, PSA level at diagnosis, biopsy Gleason score (bGS), number of positive biopsy cores, clinical stage (cT), pathological stage (pT), pathological Gleason score (pGS), positive surgical margins (PSMs), lymphovascular invasion (LVI), number of nodes removed, and positive node status (pN1). Specimen confined disease was defined as cases with negative surgical margins and no lymph node metastasis (pN0). (Page 3, Line92-94)
Comment 2:
Table 4, Figure 2, Table 5, Table 6. The notation for ≥ is incorrect. (“≧” is not correct for English literature.)
Response:
We thank the reviewer for this important observation. We have carefully revised all tables and figures, and replaced the incorrect symbol “≧” with the correct “≥” throughout (Table 4, Figure 2, Table 5, and Table 6).
Reviewer 2 Report
Comments and Suggestions for Authors
As the treatment of high risk prostate cancer has evolved to a multiple-disciplinary approach, surgery remains to be the backbone of it. Patient selection is key to determine who benefits from surgery so that over-treatment with RT or hormone treatment is avoid. This manuscript, based on 258 patients treated with surgery with a median follow up of 5 years, elegantly reported oncological (and urinary control functional) outcomes and their associated risk factors. Results of the paper will add to our knowledge base on how to better select and counsel patients.
Author Response
Comments and Suggestions for Authors
As the treatment of high risk prostate cancer has evolved to a multiple-disciplinary approach, surgery remains to be the backbone of it. Patient selection is key to determine who benefits from surgery so that over-treatment with RT or hormone treatment is avoid. This manuscript, based on 258 patients treated with surgery with a median follow up of 5 years, elegantly reported oncological (and urinary control functional) outcomes and their associated risk factors. Results of the paper will add to our knowledge base on how to better select and counsel patients.
Response
We sincerely appreciate your encouraging comments. We fully agree that surgery remains a vital component of modern multidisciplinary treatment for high-risk prostate cancer, and that careful patient selection is central to avoiding overtreatment with radiotherapy or systemic therapy. We are grateful for your positive feedback and are confident this paper will further strengthen contributions to shared decision-making and patient selection.
Reviewer 3 Report
Comments and Suggestions for Authors
The authors aimed to investigate the potential and limitations of robot-assisted radical prostatectomy (RARP) as primary treatment without perioperative systemic therapy in patients with locally advanced PCa. They relied on single center data, over a period of 10 years (2012-2022).
The authors briefly reported that 63.6% have ≥pT3a and nodal involvement (pN1) was found in 27.1%. Five-year BRFS, MFS, and CSS were 36.6%, 88.9%, and 98.3%, respectively. Treatment failure occurred in 21.3%. These percentages required a careful interpretation.
First, which kind of adjuvant therapies were excluded from the analysis? A locally-advanced PCa that not receive adjuvant RT may be undertreated and it's a detrimental result.
Second, if BRFS is 36.6% and treatment failure is 21.3%, we should question how the treatment failure was defined. Moreover, what was the management in those patients where the treatment did not fail but they did not exhibit BCR? Apparently, the criteria for defining the two outcomes are in discrepancy and they should be contextualized.
Third, according to EAU criteria, which was the BCR retrieved?
Fourth, the role of immunocompromission should be discussed. (PMID 40533295)
Author Response
We thank the reviewer for the thoughtful comments and the opportunity to clarify several important points regarding our study.
Comment 1:
First, which kind of adjuvant therapies were excluded from the analysis? A locally-advanced PCa that not receive adjuvant RT may be undertreated and it's a detrimental result.
Response
We sincerely thank the reviewer for this important comment and apologize for any lack of clarity in our original description regarding perioperative and salvage treatments.
As described in the Methods section, patients who received neoadjuvant or adjuvant systemic or radiation therapy were excluded from our analysis Thus, all included patients underwent RARP with extended pelvic lymph node dissection alone.
Regarding post-recurrence management, salvage radiotherapy (sRT) was performed in 23 patients, androgen deprivation therapy (ADT) in 75 patients, and a combination of ADT and RT in 40 patients. In addition, 16 patients who developed biochemical recurrence but had persistently low PSA levels were managed with active surveillance without immediate treatment. We have now added this information to the Results section for clarity as below.
“Among patients who developed BCR, subsequent treatments were initiated according to disease status. Salvage radiotherapy (sRT) was performed in 23 patients, androgen deprivation therapy (ADT) in 75 patients, and a combination of ADT and RT in 40 patients. Additionally, 16 patients with biochemical recurrence but persistently low PSA levels were managed with active surveillance without immediate treatment. Patients who experienced neither Persistent PSA nor BCR continued scheduled PSA surveillance.” (Page 5, Line158- Page 6, 164)
We respectfully disagree with the suggestion that omission of adjuvant RT necessarily represents undertreatment. The ARTISTIC meta-analysis, which prospectively pooled the RADICALS-RT, GETUG-AFU 17, and RAVES trials, demonstrated no significant difference in event-free survival between adjuvant RT and early salvage RT (5-year EFS: 89% vs. 88%; HR 0.95, 95% CI 0.75–1.21, p=0.70)( 13. Lancet. 2020 October 31; 396(10260): 1422–1431.) These findings indicate that immediate adjuvant RT is not mandatory in all cases, and our results should not be interpreted as detrimental due to lack of adjuvant RT. To reinforce this point, we have added a new paragraph in the Discussion section referencing the ARTISTIC meta-analysis and emphasizing that our cohort reflects a treatment approach consistent with current high-level evidence as below.
“In our study, patients who had received systemic therapy or radiotherapy as neoadjuvant or adjuvant treatment were excluded from the analysis. Therefore, all included patients underwent RARP with extended pelvic lymph node dissection alone. Consequently, all patients who received radiotherapy in this cohort under-went it as salvage radiotherapy following biochemical recurrence. The ARTISTIC meta-analysis, which prospectively integrated data from three randomized tri-als—RADICALS-RT, GETUG-AFU 17, and RAVES—demonstrated no significant difference in event-free survival between adjuvant radiotherapy and early sRT (5-year event-free survival: 89% vs. 88%; HR 0.95, 95% CI 0.75–1.21, p=0.70) 【13】. These findings indicate that immediate adjuvant RT does not necessarily provide additional benefit and support the validity of a management policy where adjuvant RT is deferred until recurrence. In our cohort, most patients who experienced recurrence received appropriate salvage therapies, while some with very low PSA levels were safely monitored without immediate intervention. Taken together, these results suggest that the omission of adjuvant RT in our study should not be regarded as undertreatment, but rather reflects a treatment approach consistent with current high-level evidence.” (Page 10, Line247-262)
We hope these clarifications and additions address the reviewer’s concerns, and we again apologize for the earlier lack of detail.
Comment 2:
Second, if BRFS is 36.6% and treatment failure is 21.3%, we should question how the treatment failure was defined. Moreover, what was the management in those patients where the treatment did not fail but they did not exhibit BCR? Apparently, the criteria for defining the two outcomes are in discrepancy and they should be contextualized.
Response
We thank the reviewer for this important point and apologize for any lack of clarity in our original wording. We agree that an explanation is needed for the apparent discrepancy between BRFS (36.6%) and “treatment failure” (21.3%). In this study, biochemical recurrence (BCR) was defined as two consecutive PSA values ≥0.2 ng/mL after the postoperative PSA had decreased below 0.1 ng/mL, which is consistent with contemporary AUA guidance【9, Morgan, T.M. J. Urol. 2024, 211(4), 509–517. 】By contrast, the outcome we previously labeled “treatment failure” refers to Persistent PSA, i.e., failure of PSA to decline to <0.1 ng/mL after surgery, a definition widely used in surgical series. 【10, Ploussard, G. Eur. Urol. Oncol. 2021, 4(2), 150–169.】These are therefore distinct endpoints: Persistent PSA reflects an immediate postoperative biochemical nonresponse, whereas BCR captures subsequent biochemical relapse among men who initially achieved a PSA <0.1 ng/mL. In our survival analyses, patients with Persistent PSA are counted as events at time zero for BRFS, while later BCR events accrue over follow-up; hence the two percentages need not match. Patients who experienced neither Persistent PSA nor BCR were managed with scheduled PSA surveillance.
Manuscript changes.
To avoid any further confusion, we will:
Replace the term “treatment failure” with “Persistent PSA” throughout the manuscript (text, tables, figure legends).
Revise the Methods:
BCR was defined as two consecutive PSA values ≥0.2 ng/mL.
→ “Biochemical recurrence (BCR) was defined as two consecutive PSA values ≥0.2 ng/mL after postoperative PSA had decreased below 0.1 ng/mL consistent with contemporary guideline definitions【9】. Persistent PSA was defined as a postoperative PSA that did not decline to <0.1 ng/mL after radical prostatectomy; for time-to-event analyses, patients with Persistent PSA were counted as events at time zero for BRFS, with the date of surgery assigned as the BCR date【10】.” (Page 3, Line108-114)
Add a brief clarifying sentence in the Results as follows:
“Among patients who developed BCR, subsequent treatments were initiated according to disease status. Salvage radiotherapy (sRT) was performed in 23 patients, androgen deprivation therapy (ADT) in 75 patients, and a combination of ADT and RT in 40 patients. Additionally, 16 patients with biochemical recurrence but persistently low PSA levels were managed with active surveillance without immediate treatment. Patients who experienced neither Persistent PSA nor BCR continued scheduled PSA surveillance. “(Page 5, Line158-Page 6, 164)
We hope these clarifications address the reviewer’s concerns.
Comment 3:
Third, according to EAU criteria, which was the BCR retrieved?
Response
We appreciate the reviewer's questions. Multiple definitions of postoperative biochemical recurrence (BCR) exist across guidelines. Because our institution's postoperative policy and interpretation of results center on early salvage radiotherapy (sRT), we adopted the definition from the AUA/ASTRO/SUO Salvage Therapy Guidelines, namely two consecutive PSA levels ≥0.2 ng/mL. 【9, Morgan, T.M. J. Urol. 2024, 211(4), 509–517.】This definition was chosen to align the endpoint with contemporary sRT decision-making and reporting standards. This threshold is broadly consistent with international practice; therefore, we believe it does not substantially alter the clinical interpretation of BCR in this study.
Comment 4
Fourth, the role of immunocompromission should be discussed. (PMID 40533295)
Response
We appreciate this suggestion. However, we consider that the role of immunocompromission is outside the scope of the present analysis, which specifically addressed the surgical outcomes of RARP without perioperative systemic therapy. Since our cohort did not involve immune status assessments or immunotherapy interventions, we believe that a detailed discussion on this aspect would not be directly applicable.
Round 2
Reviewer 1 Report
Comments and Suggestions for Authors
The article has been sufficiently revised in accordance with the reviewers' comments.
Reviewer 3 Report
Comments and Suggestions for Authors
No further comments are needed.